# Tracing the Tracts of *Qaṣaṣ*: Towards a Theory of Narrative Pedagogy in Islamic Education

Muhammad Fawwaz Bin Muhammad Yusoff

Faculty of Quranic and Sunnah Studies, Universiti Sains Islam Malaysia, Nilai 71800, Negeri Sembilan, Malaysia; fawwaz@usim.edu.my

**Abstract:** The concept of narrative holds a pivotal position in the Qurʾān, yet it has been subject to inadequate scrutiny and insufficient representation in pedagogical discourse concerning Islamic education. The present work endeavours to rectify this gap in knowledge by employing the technique of constructivist grounded theory to the Qurʾān and major exegeses, with a particular focus on the term *qaṣaṣ*, which pertains to the notion of narrative. This article delves into the profound tracts and *maqāṣid* (objectives) that *qaṣaṣ* hold in the Qurʾān and contemplates their exhortation for education on Islam and modern pedagogy. The analysis reveals that the *qaṣaṣ* present in the Qurʾān serves as a fundamental framework that directs the essence of the narrative pedagogy model of teaching and learning between the pedagogue and learner. Through typological figuration, the listener's contemplation leads to a re-evaluation of conventional notions surrounding the dynamics between teacher and student and the dissemination of narrative within a pedagogical setting. The triad of truth, beauty and explication are fundamental pillars within this Islamic framework for narrative pedagogy, representing the essence of the human condition concerning education. Because these domains emerge from the concept of *qaṣaṣ*, the integration of the framework into Islamic education is a matter of utmost importance, given its centrality in the Qurʾān to foster and perfect the principles of Muslims and their sense of self.

**Keywords:** al-Qurʾān; *qaṣaṣ*; *maqāṣid*; *Qiṣāṣ al-Anbiyā*; narrative pedagogy; Islamic education; typological figuration; *mimesis*

## 1. Introduction

In a society where information is readily available, imparting knowledge and wisdom from a virtuous being to the masses, as exemplified in the Islamic intellectual tradition, may appear to be a regressive and contradictory step. Amidst the multitude of signifiers that generate diverse contexts, contemporary knowledge practises appear to be bound by the chains of overstimulation, particularly through technology. In contrast, the Islamic tradition has preserved a distinct system that strives to uphold its connection to the revered Prophet Muḥammad and earlier prophets. What is the significance of meticulously selecting and fixating on the establishment of these connections? One may ponder how the former can offer insight into the latter's dissonances in contemplating the intersection of premodern knowledge transmission and modern comprehension. The concern for pedagogy, or the method and practice of teaching, has been present since the classical Muslim community. What has stood out from the earliest centuries is that narrativisation has played an essential role in Islamic teaching, particularly spiritual, theological and ethical learning (Lawson 2022). Hayden White (1987) and Monika Fludernik (1996) posit that narrativisation involves imbuing a discourse with the narrative structure to enhance comprehension of the depicted phenomena.

Recently in his ground-breaking study, Cyrus Zargar (2017) demonstrated the use of narrative to transmit ethical, intellectual, and spiritual principles, which are exceptionally frequent in Islamic civilisation. He asserts that narrative connects Sufi and philosophical

virtue ethics better than any other common threads. Mohammed Rustom's (2020) "*Storytelling as Philosophical Pedagogy: The Case of Suhrawardī (d. 587/1191)*" provides a close reading of Suhrawardī's *Āwāz-i parri Jibrāīl* (The reverberation of Gabriel's wing) in order to demonstrate the text's employment of symbolic language in conveying its educational communication. In "*Muhammad as Educator, Islam as Enlightenment, and the Qurān as Sacred Epic,*" Todd Lawson (2020) sheds light on the literary construction of the Qurān as well as its themes of knowledge and education. The study synthesises an exploration of the appearance of epic forms and themes in the Qurān with a discussion of Prophet Muhammad's heroic role as the educator of humanity. While Nadja Germann's (2020) "*How do We Learn? Al-Fārābī's Epistemology of Teaching*" scrutinises an understudied dimension of al-Fārābī's works on education, his epistemology of teaching. Interestingly, she describes al-Fārābī's recognition of teaching as the transmission of an intellectual heritage that consists of an evident corpus of antique and late-antique texts. Moreover, a decent number of Arabic works before and after al-Ghazālī (505/1111) also deal in a most captivating way with various aspects of didactics and pedagogy (Günther 2005).

Each attempt to systematise a body of knowledge necessitates preserving, transmitting, and cultivating those efforts (Muhammad Yusoff 2019). Undoubtedly, scholarly discussions on theoretical and practical concerns in teaching and learning or pedagogy are significant components of various Islamic works from the classical and modern periods. Philosophers, theologians, jurists, *ḥadīth* experts, litterateurs, and natural scientists were among the Muslim scholars who addressed pedagogical and didactic ethos. To some extent, the question of pedagogy is the product of the various orientations and opinions among scholars who addressed this topic. According to Muhammad Hamidullah (2003), the concept that ethical behaviour could be found in the great individuals of the past prevailed not just in the Qurān and pre-Islamic Arabian traditions but also in the *ḥadīth*, stories of companions, and other accounts that functioned as Qurānic exegesis. Stories of ancient prophets, such as those described in Ahmad ibn Muhammad al-Thaʿlabī's (d. 427/1036) *ʿĀraʿis al-Majālis fī Qiṣāṣ al-Anbiyā*, provided standards of behaviour while also reinforcing the truth of the Qurān, which references to facts in the lives of those prophets (Zargar 2017). More than any other component of Islamic life, the Qurān is at the centre of Islamic history and culture, holding power, inspiring conceptions, conduct, definitions, visions, and the literature. Every genre of the Islamic literature is thus, to varying degrees, the result of an acquired and refined Qurānic legacy. Whether in poetry or legal cases, the Qurān pervades practically every page (Tottoli 2017).

## 2. Materials and Methods

Adjacent to the backdrop of these intellectual traditions, there is a need for a theoretical framework that allows Muslims to explore the purposeful complexity of the roles that narratives play in their lives. In short, I endeavour to bring to the forefront the narrative pedagogy of Islamic intellectual tradition by redressing the tracts, purposes, or *maqāṣid* (objectives) of the Qurān. Haverkamp and Young (2007) asserted that research methodologies are placed within epistemological and ontological perspectives and the philosophy of science. The Islamic paradigm has its epistemology and ontology, which may seem interpretive or constructivist from the outside. Indeed, the lived reality of Islam, as embodied in the life of Prophet Muhammad (PBUH) and transmitted through the scholarly tradition of the Muslim ummah (global community) (Muhammad Yusoff 2020), can undoubtedly be said to incorporate constructivist approaches to social phenomena as Muslims grapple to make sense of how to apply religious tenets to daily life (Al-Sharaf 2013). As can be seen, this is represented in the different *fawāriq* or *madhāhib* (schools of thought) equally acceptable within the tradition and the historical precedence for diverse interpretations and viewpoints of Islamic principles in practice (Al-ʿAlwānī 1993). This dimension of the phenomena, i.e., narrative pedagogy under investigation in this study, includes a range of Islamic scholarly voices in developing a theory of narrativity and using constructivist grounded theory as the methodology that comprises the more prominent

research. Individuals who have devoted their lives to studying and comprehending the meanings and knowledge contained in Islamic sources, as well as those who work closely with the Muslim community and possess knowledge about the relationship between Islam and education, hold the keys to information and insight gained through their unique experiences (Rothman 2022).

Essentially, an educator may view the Qurān or *ḥadīth* and its commentary as a source to construct a didactic ethos. In comparison, an exegete may view it as a scriptural base to postulate a concept over ethical, legal, or religious principles that spans generations of scholars. These two perspectives are not seen as mutually exclusive in this paper. On these premises, this article offers insights into specific Qurānic terms and their interpretations, which due to their scholarly originality, are actual landmarks in the history of Islamic education. A systematic approach to developing a framework anchored in these sources is required to properly place the area of narrative pedagogy in Islamic education and pave the way for future research on indigenous intervention development. Therefore, to contextualise the discourse over the intersection between narrative, pedagogy and the meaning of the Qurānic term, the article delves into the general conceptualisation of narrative pedagogy, early Muslim views on developing didactic method, and Qurānic term on narrative by emphasising some of its distinguishing aspects and some interpretative deliberations correlated with Qurānic exegeses.

Firstly, we will investigate the topic of narrative pedagogy or storytelling in a broad sense and draw a map of it. Being exposed to the significance of storytelling in the writings of classical scholars, elaborating on the question of didactic or pedagogy brings a reminder of the depiction of narrative in the Qurān. In the first phase of any Muslim studies, the question underpinning the research is, "What are the fundamental principles and notions that govern the conceptualisation of the narrative pedagogy deduced from the Qurān?" Although there are many possible interpretations and viewpoints on what the Qurān says about the narrative pedagogy and how this can be applied to formulate a structure or map of the relationship, it was found that many common foundational factors (Abdel Haleem 2017) emerged as the key in conceptualising a general framework of narrative pedagogy (Stelzer 2016). The question of terms in the Qurān towards narrative pedagogy must be addressed from the perspectives of notable exegetes. Likewise, I propose narrative pedagogy as a predicate logic that refers to knowledge representation mentioned in the Qurān and includes both general terms and narration terms of a particular didactic, specific verb in connection with the general narrative terminology, and universal reflective lesson, which pertain to every dimension of the human experience. Therefore, in the last part, I will focus only on a Qurānic term, object or noun that is *qaṣaṣ* for depicting the purpose, understanding and deliverance of narrative pedagogy, and it explains why the term is the scope of narrative pedagogy. When it comes to narration, the Qurān often uses the verb form *qaṣṣa* to relate stories and histories, while the noun form *qaṣaṣ* is used to denote an account or report (Badawi and Muhammad 2008).

## 3. Studies of the Qurānic Literature and Narrative Pedagogy

Striking a holistic balance between various levels of interdisciplinarity will continue to be a significant challenge for anyone seeking to make formative contributions to the subject of Islamic education in any area. Understanding the many forms of interdisciplinary methods and how they vary from disciplinary approaches provides a fuller understanding of the process of knowledge generation and transfer (Bin Sapi'ee and Muhammad Yusoff 2022). The relation of narrative and pedagogy has broadened the scope of the education in which Muslims and the Qurān must be addressed. Therefore, it is impossible to unthinkingly embrace or eliminate educational viewpoints developed outside the Islamic tradition and explain their alleged "incongruity" with Islam by invoking passages from the Islamic scriptural foundation, such as the Qurān and Sunnah. On the other hand, it is difficult to justify the methodologically approach of didactic issues raised by contemporary educationists developed outside the Islamic tradition only through the lens of *fiqh* (Bin

Muhammad Yusoff 2022). However, this is not to say that all these levels and aspects of interdisciplinarity should always be included in the Islamic discourse on any pedagogical topic. Hence, in what follows, I attempt to review the practice of narrative as pedagogy which was far more than just narrating stories. A limited treatment also tries to identify the nuances related to such a practice in the light of broader cultural milieus. These sketchy indications should be sufficient to realise the importance of narrative, narrativity, narrativisation, pedagogy, etc.

### 3.1. Narrative as Pedagogy

Despite its significance, the discourse surrounding the narrative form must pay more attention to the pedagogic encounter. The act of introspectively contemplating and analysing the stories that shape one's existence during the educational journey defines the concept of narrative pedagogy (Goodson and Gill 2011). Goodson argues further that the philosophy of narrative pedagogy offers a valuable tool for elucidating the narrative mechanisms that facilitate profound transformation and growth for both individuals and collectives in educational settings and beyond. Over time, a burgeoning collection of written works have come to acknowledge the potency of storytelling in eliciting shifts in belief, furnishing the framework for profound metamorphoses, and conveying knowledge. As we delve into the classical writings of narrative, we come across instances that resemble the initial conception of the Islamic literature.

In the realm of philosophical discourse, it is widely acknowledged that in his wisdom, Plato crafted dialogues to convey his ideas. These dialogues were characterised by a narrative structure, wherein arguments are seamlessly interwoven within the fabric of the story (Scott 2007). To deal with this topic, one must consider the concept of *mimesis*, derived from the Greek term for imitation or representation, which has served as a fundamental framework in the realm of Western art theory, dating back to the era of Greek antiquity. Plato's philosophical framework regarding representation is rooted in a fundamental dichotomy between *mimesis* and *diegesis*. The concept of *mimesis* has been a subject of profound philosophical inquiry since ancient times, as evidenced by the seminal works of Plato in his *Dialogues* (Republic, Bks. III and X; The Sophist; La) and Aristotle's *Poetics*, which followed shortly thereafter. The divergent conceptions put forth by these two philosophers have played a pivotal role in shaping the Western mindset with regard to *mimesis* and fiction.

Bowery (2007) notes that Socrates provides inferential justifications for philosophical beliefs such as anamnesis and the immortality of the soul. Examining discourse, one may contemplate the interplay between myth, allegory, and other narrative elements such as the Cave, the Divided Line, the Myth of Er, and the Chariot. The notion that arguments in the inferential sense or their theoretical conclusions could be deemed the fundamental unit in the Platonic literature is far from self-evident. It is intriguing to observe the correlation between the philosophical musings of Hume, Berkeley, Malebranche, and Diderot and the manner in which they articulate their thoughts through the medium of conversation (Harrelson 2012). The act of publishing philosophy in biography has been a common practice among notable philosophers such as Plutarch and Laertius. On the other hand, Schopenhauer recommended the practice of philosophy in autobiography but did not personally engage in it.

Meanwhile, Augustine, Rousseau, Nietzsche, and Collingwood have all penned their philosophical musings in the form of an autobiography (Mathien and Wright 2005). The narrative has not only been a crucial element in the literary forms that philosophy has assumed but has also been recognised for its importance in the structure of reason. The pinnacle of anglophone philosophy occurred in modern times, as Alasdair MacIntyre (2017) posited in *After Virtue* that the essence of humanity lies in our innate ability to craft and share stories. Through narrative, human beings unlock the secrets of personal identity, delve into the complexities of human behaviour, and explore the timeless quandaries that have long perplexed them. He contended that the crux of personal identity, the scrutiny

of conduct, and other customary themes could be comprehended through the narrative lens. In due course, Paul Ricoeur delved deeper into these aspects, dedicating his later endeavours to the contemplation of narrative and its significance in the realm of philosophy, including the philosophy of history and the metaphysical implications of time (Joy 1997). As Ricoeur ([1986] 1991) puts it, hermeneutics represents a philosophy deeply rooted in the concept of perpetual mediatedness. It underscores that no form of self-understanding exists devoid of mediation through signs, symbols, and texts.

The aforementioned points pertain solely to the narrative manifestation of the profound bond between certain philosophers and the narrative. The interplay between narratives and inferential arguments is crucial, as it speaks to the very nature of human reasoning and communication. While there may be distinct disparities between these modes of expression, their ability to coalesce and form a cohesive whole is a testament to the complexity and nuance of our intellectual faculties. Nevertheless, one also finds that to understand narrative pedagogy, it is necessary to consider the "narrative" constructing the framework and the actual term of teaching or pedagogy. How can we teach and learn with narrative orientation? In social research, narratives and stories are sometimes used interchangeably. Hinchman and Hinchman (1997) assert,

> Narratives (stories) in the human sciences should be defined provisionally as discourses with a clear sequential order that connect events in a meaningful way for a definite audience and thus offer insights about the world and/or people's experiences of it.

Clearly, Hinchman and Hinchman's definition and characterisation of narratives stress the significance of social interaction in constructing narratives and transforming human experience into meaning. According to Rogers (2011), narrative pedagogy is "any educational method that intentionally uses the narrative arts of storytelling, drama, and creative writing to nurture theological reflection, spiritual growth, and social empowerment." Hence, narrative pedagogy should be discerned from narrative method, as the former is deliberate strategy by a teacher or educator to nurture theology and spirituality which has a mimetic function, whereas the latter constitutes temporality; voice (who speaks?); vision or focalisation (who perceives?); and style (James and Booth 2007).

### 3.2. Narrative Pedagogy in the Islamic Intellectual Tradition: A Selection of Approaches

Al-Attas (2018) depicts the sages, men of discernment and the learned scholars among the Muslims of earlier times who combined *ʿilm* with *ʿamal* and *adab* and conceived their harmonious combination as education. He further asserted, "Education is in fact *taʾdīb*, for *adab* as here defined already involves both *ʿilm* and *ʿamal*." Survey the classical and contemporary literature education in the Muslim world, and we quickly discover that discreet terms emerges: *taʿlīm* (instruction), *tadrīs* (the delivery of a lecture), *taʾdīb* (inculcation of virtues), and *tarbiyah* (nurture). It must be noted that a considerable body of the literature explores narrativity or the literature as a method and medium of instruction (Stelzer 2016; Zargar 2017; Lawson 2022). Throughout the vast expanse of Islamic lands from the early to the contemporary era, one can find a plethora of proverbs, aphorisms, and wise sayings that delve into the content, objectives, and details of Islamic education. These timeless expressions encapsulate the wisdom and knowledge that have been passed down through generations of scholars and thinkers. It goes without saying that narrativity plays a significant role in Islamic intellectual tradition, particularly in transmitting knowledge and communicating Islamic theology.

Armstrong (2016) seeks to build a detailed portrait of the society in the early Islamic period, up to the year 750 AD or to the late Umayyad period, in which he attempts to answer questions about the identity of *quṣṣāṣ* (storytellers), their origins and functions, their links with dominant religious and political currents, and even the nature of their *qaṣaṣ*, which he classifies as religious, martial, or religio-political. He describes in detail their abilities and conduct, as well as the extent of their knowledge and oratory skills, which included their *lisān* (linguistic abilities), *bayān* (rhetorical skills), and *ʿilm* (religious knowledge), before

attempting to analyse their role, whether they were considered *aṣḥāb bidʿah* (innovators) or religious conformists. Although Amstrong conducted extensive research and made various attempts to analyse and reconcile the contradictions of the concepts of *qaṣaṣ* and *quṣṣāṣ*, the final image of what truly constitutes a *qāṣṣ* (storyteller) and a *qiṣṣah* (story) remains unclear. It seems that the answer to this question is still undefined. His writings more like a lengthy historical account rather than an analytical exploration of the term itself, especially from the Qurʾān or ḥadīth, which could be attributed to the evolving and ambiguous nature of the available sources. Meanwhile, Amir's (2022) discourse delves into the intricate realm of early Islamic ethics and the transmission of knowledge as it pertains to the premodern dichotomy of *ʿulamā-quṣṣāṣ*. He posits that the dynamic between the *qāṣṣ-ʿālim* (preacher–scholar) involved a philosophical methodology rather than a contentious one that gave rise to both accurate and erroneous schools of thought. The phenomenon at hand can be viewed through a philosophical lens as a grand premodern exercise in *jadal* (argumentation) and *munāẓara* (debate), which unfolded gradually within the Islamic tradition. It was not necessarily concerned with the concepts of rupture or continuity, as these are often attributed to modern historians.

The distinctions, or rather the criteria for differentiating narratives, are instructive. According to al-Ghazālī (d. 1111 AD), "knowledge of the stories (narrated) in the Qurʾān" is at the bottom and apex of the scale of sciences of the Qurʾān as he defined in his The Jewels of the Qurʾān (Quasem 2018). For al-Ghazālī, "knowledge" ultimately signifies a "knowledge of God", and this knowledge has a distinct nature. As a result, one must infer that "the knowledge of the stories" is not only the furthest away from God's knowledge because it delivers just the weakest glimpse but also because it conforms least to the concept of knowledge that led to such a categorisation system. The *ʿulūm* (knowledge) that follow "the knowledge of the stories," according to al-Ghazālī, include *ʿilm al-kalām* (theology), *fiqh* (jurisprudence), and continuing all the way to the "knowledge of God," and appear to be built on the concept of knowledge familiar from the so-called exoteric or 'rationalist' sciences (Lumbard 2019). As a result of this categorisation, al-Ghazālī does not consider persons who narrate stories to have accomplished major intellectual accomplishments. This is noteworthy in and of itself because it highlights the vital relationship between storytelling and knowledge, and the reality that the significance given to tales for the transfer of knowledge is heavily dependent on our ideas of knowledge.

Among Ibn Sīnā's numerous works, his stories (*qiṣāṣ*) or autobiography have a very distinctive place by virtue of their form and of their content (Stroumsa 1992). Apart from the biography written by his student and amanuensis al-Juzjānī, his autobiography is set within a presentation of the Aristotelian curriculum, which Ibn Sīnā deemed essential to the philosophy of his times (Reisman 2013). According to Dimitri Gutas (2014), the stories serve as an illustration of the symbolical method or allegorical communications employed by the Aristotelian philosopher. He added that "the allegorical method of communication is inferior to the demonstrative and expository" because it is suited for and addressed to inferior minds. In *al-Ishārāt wa al-Tanbīhāt* (Remarks and Admonitions), his last work on metaphysics, Ibn Sīnā alludes,

> Those who possess divine knowledge vary in their rank and level, and this sets them apart from others even in their terrestrial existence. It is as if they wear their bodies like cloaks, which they later shed, turning towards the realm of sainthood. These individuals possess both covert and manifest matters, with the ignorant often disapproving of the former while those who are knowledgeable cherish them. We shall disclose such matters to you (*naquṣṣuhā*). Therefore, if among the things that reach your ear and the tales you hear, there is the story (*qiṣṣah*) of Salāmān and Absāl, know that Salāmān is an image (*mathal*) referring to yourself and that Absāl is an image referring to the stage you [have reached] in Knowledge if you are worthy of it. Now follow the hint (*ramz*), if you can. (Inati 2014)

However, Stroumsa (1992) demonstrated that these stories do not fit the symbolic method used by the Aristotelian philosophers; it is nonetheless important to understand

them in light of Ibn Sīnā's interpretation of Aristotelian philosophy. For Corbin (1988), he rejected the designations of mystical allegories or philosophical narratives or tales for a reading of Ibn Sīnā's writings. Instead, he chose "recitals or visionary or initiatory recitals". Perhaps he realised that the "recital" had a mimetic role, which is very close to grasping the precise meaning of the *qiṣṣah*.

### 3.3. The Qurān and the Narrative Literature

It is comparatively important to recognise that a significant portion of these educational considerations or methods bears the imprint of ancient Greek *paideia* ('rearing', 'education'), which primarily revolved around philosophical ideas (Günther 2016), and Islamic educational discourse was profoundly informed and moulded by religious principles. In the opinion of Todd Lawson (2022), narrativity plays a significant role in the Qurānic kerygma or message. The narratives found in the Qurān typically have a beginning, middle, and end. His study on Qurānic Kerygma compares and contrasts Northrop Frye's understanding of kerygma in the Bible. In his three major studies of the Bible, Frye relies heavily on an understanding of kerygma as the proclamatory rhetoric of the Bible. Similarly, Lawson explores the storytelling and preaching elements of the Qurān's message or kerygma. However, Lawson emphasises that the character of the Qurānic kerygma is unique in the world literature, though it may have some features in common with the Bible.

In a broad sense, the classical to modern corpus addressing matters of narrativity or storytelling were characterised by their strong foundation derived from the Qurān and the ḥadīth or traditions of the Prophet Muhammad (PBUH) (Al-Khaṭīb 1975). The profound influence of the Qurān has given rise to a rich body of the educational, creative, and historical literature centred around the narratives of the Prophets and sages (Dutton 2020). Thus, the literary tradition encompasses various genres prevalent in Islamic civilisation, serving as a platform for exegetical endeavours, narrative embellishments, and theological explorations (Rustom 2021). While this process of literary development continues to unfold in the present day, its formative moments can be traced back to the early centuries of Islam. It is a longstanding belief that every piece of the Islamic literature can be traced back to a direct or indirect interpretation of the divine text or, at the very least, has a clear connection to it, whether overt or covert. The intricate relationship between the Qurān and subsequent literary works is inherently nuanced and multifarious. The Qurān possesses the potential to not only mould literary dialogue straightforwardly but also serves as an autonomous frame of reference to rejuvenate other notions and musings. Much like other religious and literary traditions, the Islamic literature showcases various customs that can be categorised into three types - those derived from the holy book through direct and rational means, those that align with its essence, and those completely unrelated to it (Bin Muhammad Yusoff and Ismail 2023).

The narratives found within the Qurān possess a distinct historical context and are intricately linked to three specific categories or temporal phases in history: the past, the present, and the future. Bennabi's (2004) insights effectively encapsulate this perspective regarding narratives within the Qurān. Focusing on the narrative literature, Roberto Tottoli (2017) retraces literary legacy and how it has been influenced, or in other words, the mark left by the Qurānic narratives on the Islamic Arabic literature. The relationship between the Qurān and the subsequent literature is necessarily composite, and Tottoli evaluates the Qurān on the one hand and the religious literature on the other to identify where the former has influenced the latter or where the latter has absorbed elements of the former. In his exploration, he delves into the vast expanse of the Qurānic literature, encompassing diverse subjects such as pre-Islamic history, the life of the Prophets, and the eschatological future. This segment of the narrative literature contains all of the distinctive typologies in the relationship between sacred text and non-canonical tradition (Shah 2013). In the case of the literature about the prophets, the Qurān influences the delineation of themes and fundamental roles, providing a narrative outline and frequent references to specific events. A notable development swiftly took place as a chronological framework

was established in the Islamic literature, spanning from the beginning of creation to the impending arrival of Prophet Muhammad. The clearest example is Ibn Kathīr (d. 774/1373), with his follow-up to his universal historiography *al-Bidāyah wa al-Nihāyah* (The Beginning and the End). Medieval scholars like Ibn Kathīr assemble diverse stories and traditions in this temporal and literary domain, interweaving them with Qur'ānic verses. Within these works, the Islamic tradition sought to merge a cohesive narrative incorporating the Qur'ānic principles and historical accounts into a comprehensive and chronologically ordered account of prophetic stories. This phenomenon primarily takes place within the realm of historiography, particularly in historical texts and the earliest works specifically devoted to recounting the lives of the prophets, known as *Qiṣāṣ al-Anbiyā'*. Among the famous scholars contributing to the genre of the *Qiṣāṣ al-Anbiyā'* are Wahb b. Munabbih (206/821), Abu Jaʿfar Muḥammad b. Jarīr al-Ṭabarī (310/923), Abu Isḥāq al-Thaʿlabī, and many others (Brinner 2002).

Nevertheless, the relationship between the narrative material of *Qiṣāṣ al-Anbiyā'* and the Qur'ān is characterised by complete autonomy, as these stories are imposed upon the Qur'ānic corpus. This distinction is even more apparent in a particular genre that combines passages from the Qur'ān with the ḥadīth, known as the literature on the *asbāb al-nuzūl* (occasions of revelation). In this genre, an endeavour is made to assign dates to the revelation of specific Qur'ānic verses by linking them to events in the life of the Prophet Muhammad. From a literary perspective, these considerations hold little significance. The Qur'ān, in its narrative function, does not delve into the life of Prophet Muhammad, necessitating the construction biography of the Prophet using sources independent of the literary dimension of the Qur'ān. It will suffice to provide an example from the most well-known biography of Muhammad, *al-Sīrah al-Nabawiyyah*, composed by Ibn Hishām (d. 218/833) from the original text by Ibn Isḥāq (150/767), to demonstrate the point. Nevertheless, from the earliest times, the *Sīrah* and other writings offer a cohesive manifestation of the core essence of Islam, encompassing both individual religious devotion and spirituality, as well as the broader scope of the communal religious existence of humanity (Azmi 2022).

There is no space here to elaborate in detail, but it is important to observe that at the heart of these writings lies the concept of transformation through recognition and education. They entail acknowledging the spiritual and intellectual poverty caused by the existing condition and recognising the Prophet, who will lead Muslims away from such a state. Here, Lawson (2021) proposes typological figuration in reading the stories or 'epic', in his term, of the Qur'ān. Lawson draws attention to the role of typological figuration as "it connects the historical process of revelation, extends it to all humanity and renews, authenticates and enlivens it ever time the Qur'ān is recited or read." In its narrowest sense, typological allusion involves relating characters and events to the life of Prophet Muhammad, prophets in the Qur'ān or subsequent figures of religious authority, e.g., four caliphs, companions, etc. However, this technique was more intricate in practice, encompassing the interpretation of history and the creation of narratives that emphasised connections between worldly events and events in the Qur'ān. As a result, the world itself was assimilated into the sacred text. Thus, typological figuration and pervasive and characteristic literary structures within the narrative effectively communicate its message and form to an extensive readership and audience that extends well beyond the confines of the Arabic language.

## 4. *Qaṣaṣ* and Narrative Pedagogy: An Appraisal from the Qur'ānic Exegeses

Ibn al-ʿArabī once said that one should continue to recite Qur'ānic verse until one reaches a state in which one feels as if the Qur'ān were being revealed to the reciter at the moment of recitation (Nasr 2007; Chittick 2021). As with many other subjects, evaluating the content of the Qur'ān will not provide straightforward solutions to the concerns that the study of narrative pedagogy attempts to address. Thus, mapping an Islamic narrative pedagogy viewpoint requires additional intellectual and academic work led by a set of procedures and ideas that evolved throughout the course of Islamic history. What is

required is a far more fundamental analysis that considers the Qurʾān's deliberation along with the subtlety of Muslims' lived experiences, and the unique characteristics of its community, etc. (Sulayman 2013).

Moreover, in the writing that follows, it will be argued that *qaṣaṣ* in the Qurʾān, with the core meaning of "story" or "narrative", could be a possible term, given that its semantics, functions and interpretations provide the fundamental construction of both narrative and pedagogy in Islamic education. It is also interesting to trace the term narrative, which has several Arabic words employed within the semantic field like *ḥadīth* (denotes primarily a saying or an account of action of the Prophet, but also means "narrative," "speech,", etc.), *sīrah* (lit. "way of acting," it is also used for "battles", "story", or "biography"), khabar (information, statement, narrative, or piece of history), *riwāyah* (the verb means to recite, or transmit a story or a poem; thus, a transmission or version; nowadays, a novel), *athar* (story, tradition), *ḥikāyah* (the verb from which it derives means "to relate,"; thus, "narrative," "story"), *samar* (literary entertainment, mostly at night), *khurāfah* (incredible tale, legend), *ustūrah* (legend, history without foundation; in present usage, sometimes also a myth), *nādirah* (short, witty, subtle and amusing anecdote), *mathal* (parable), and *maqāma* (assembly) (Gilliot 2006).

The Arabic root *q-ṣ-ṣ* (*qaf- ṣād- ṣād*) has a variety of meanings and applications, all of which are woven into the fabric of the Qurʾān. At its core, *q-ṣ-ṣ* implies a sense of cutting, clipping, or shearing and is associated with scissors, chips, cuttings, and even breastbones (Ibn Fāris 1979). However, this root also has connotations of retaliation and reprisal, as well as storytelling and narrative. In the Qurʾān, the root *q-ṣ-ṣ* appears in three different forms a total of 30 times: 20 times as the form I verb *qaṣṣa* (to narrate), 6 times as the form noun *qaṣaṣ*, and *qiṣaṣ* 4 times (https://corpus.Qurʾān.com/Qurʾāndictionary.jsp?q=qSS, accessed on 27 January 2023). These forms convey a range of meanings, including narration, mention, declaration, clarification, and tracking. When it comes to narration, the Qurʾān often uses the form *qaṣṣ* to relate stories and histories, while the form *qaṣaṣ* is used to denote an account or report (Badawi and Muhammad 2008). One of the meanings of *qaṣṣa* is to narrate or relate a story or history. Through narration, one can clarify and explain the meaning of the text, as well as track and follow events that occurred in the past. This process of relating a story is important in understanding the Qurʾān, as it sheds light on the context and circumstances in which the text was revealed. Significantly, the Qurʾān *sui generis* contains many stories that serve to teach and inspire its readers. These stories often illustrate important moral and ethical principles and are a testament to the power of storytelling in human culture. In addition to storytelling, *q-ṣ-ṣ* also encompasses the concept of retribution and just punishment, which is known as *qiṣāṣ* in Arabic. This form of *q-ṣ-ṣ* appears in the Qurʾān 4 times as a reminder that just retribution is necessary to maintain a fair and balanced society. The concept of *qiṣāṣ* is closely tied to the law governing acts of homicide, and the Qurʾān calls for just retribution in such cases (Badawi and Muhammad 2008).

In light of the above, central to the entire representation of narrative pedagogy is the noun *qaṣaṣ* with its derivatives that appear in the Qurʾān. After multiple readings and our humble understanding of the Qurʾānic verses, in which reference was made to *qaṣaṣ* in relation to verses that came before and after them, *asbāb al-nuzūl* (occasions of revelation), and the surah in which they appeared, as well as interpretations by prominent exegetes, we propose a framework of narrative pedagogy consisting of three domains of narrative pedagogical relationships: (1) Narrative and Truth (Q3:62); (2) Narrative and Beauty (Q12:3); and (3) Narrative and Explication (Q7:176, Q12:111, Q18:64, Q28:25). This *qaṣaṣ* word is used here as a noun in the restricted, technical sense, to refer broadly to the narrativity categories. In the context of the *asbāb*, the precise or approximate dating of the revelation of a particular verse holds little significance for the purpose of this paper. Instead, the focal point of analysis lies in understanding the context surrounding the incident described within the verse. The confirmation and expansion of categorisation of narrativity verses, as identified by previous interpretations of Qurʾānic exegetes and studies, speak to the ongoing exploration of the nature of narrative and their significance in

Muslim's understanding of the narrative pedagogy. Muḥammad Ibn ʿĀshūr (1984) delved into the intricacies of narrative in the Qurʾān, highlighting three distinct categories that encompass the relationship between narrative and guidance. These categories include narrative with explication, narrative and eternal truth, and narrative that delivers lessons from history. Ibn ʿĀshūr (1984, 1/65) alludes,

> One of the adab aspects of the Sharia is the knowledge of the history of its predecessors in legislation, starting from the prophets and their respective laws. Thus, the inclusion of narratives about the prophets in the Qurʾān serves as an embellishment of the lofty status of Islamic legislation. By mentioning the history of the legislators, Allah says, "Those unto who We have given the Book recognise it as they recognise their children, but a group of them knowingly conceal the truth [The Qurʾān 2:146].".

Similarly, Abdel Haleem (2017) expounded on the concept of narrative in the Qurʾān, identifying three interconnected ideas that revolve around the relationship between narrative and truth, narrative and context, and narrative and facts. In short, examining the three narrative pedagogy relationships in the present study follows the conventional order of the *muṣḥaf*. Against this background, this endeavour delves into the interpretative nature of these relationships.

### 4.1. Narrative and Truth

According to Ibn ʿĀshūr (1984), Qurʾānic narrative is exceptional not only because it is lesson-based and restricts itself to true stories but also because of its efficacy in carving its timeless lessons into the worldview of its readers and steadily strengthening their interest in imbibing them (Mebrouk and Zaid 2021). Let us now explore the reason for the particular affinity between 'narrative' and 'truth' in the literary context of exegeses in order to determine the pedagogical functions in the stories revealed in the Qurʾān. The first verse in the Qurʾān that mentions *qaṣaṣ* is in the third chapter, Sūrah Āl ʿImrān; verse 62 emphasises the description as well as a theoretical and practical exposition of the truth contained in *al-qaṣaṣ al-ḥaq*.

> This is indeed the true account; there is no god but God, and truly God is the mighty, the Wise. (3:62)

For a comparative diction, *al-qaṣaṣ al-ḥaq* is rendered differently in English translations. For instance, Pickthall translates it as 'the true narrative'; Abdullah Yusuf Ali renders it as 'the true account'; Sahih International as 'the true narration'; Muhammad Asad (2008), 'the truth of the matter'; and A. J. Arberry, 'the true story' (https://corpus.Qurʾān.com/translation.jsp?chapter=3&verse=62, accessed on 27 January 2023). While all the translators choose 'true' for the nominative masculine adjective *al-ḥaq*, it is obvious that there are several distinct dictions for *al-qaṣaṣ*. The term *ḥaqīqah* (reality), which is derived from it (Bin Jamil 2022), refers at the same time to the Truth and to truth in whatever context and at any level of reality with which one is concerned. In the same way that the word realisation contains the term real, spiritual realisation in *taṣawwuf* is called *taḥaqquq* (from the word *ḥaqq*), and it is well known that the accomplished Sufi is called *muḥaqqiq*. As the typical structure of the Qurʾān using the concept *munāsabah* (coherence/organic unity), most exegetes state this verse 62 of surah *Sūrah Āl ʿImrān* is preceded by three related verses (Al-Ṭabarī 2001; Al-Rāzī 1981; Al-Qurṭubī 2003).

According to *ḥadīth* reports of *asbāb al-nuzūl*, these verses were revealed during the visit of a Christian delegation from Najrān to Madīnah (Al-Ṭabarī 2001). This delegation argued that since Jesus was born without a human father, he must be considered the true son of God. Verse (Q3:59) presents one of the central arguments in the Qurʾān against the divinity of Christ. While it acknowledges the miraculous nature of Jesus' birth, it rejects the notion that it makes him divine. According to Al-Qushayrī (2000), this verse is evidence of the analogy's validity. The comparison between Jesus and Adam was based on the fact that, like Adam, Jesus was created without a father, not that he was created from dust.

The Qurʾān contends that if God could create Adam from dust without a father or mother, he could similarly create Jesus from the "blood" of Mary, as argued by Al-Ṭabarī (2001). Meanwhile, Al-Rāzī (d. 606/1210) added that the use of the word *mathal* (likeness) in Q3:59 indicates that certain attributes of Adam and Jesus are similar. The Divine command *kun* ("Be!") in the same verse appears in multiple other verses in the Qurʾān, including Q2:117, Q6:73, Q16:40, Q36:82, and Q40:68. Furthermore, verse Q3:60 warns against doubting the depiction of Jesus presented in the previous verse, as per al-Ṭabarī.

Al-Rāzī (1981) and Al-Qurṭubī (2003) highlight the phrase *al-qaṣaṣ al-ḥaqq* ("the true account") in this context can be interpreted in two ways. Firstly, it can be understood as a reference to the preceding verses that accurately depict the reality of Jesus and the events that led to the prayer challenge. Alternatively, it can be viewed as a testament to the veracity of the Qurʾān as a whole and the accounts it conveys. Al-Rāzī adds further that *al-qaṣaṣ al-ḥaqq* refers to "the collected discourse encompasses what guides towards religion, directs towards the truth, and commands the pursuit of salvation. Allah revealed to His Prophet *al-qaṣaṣ al-ḥaqq* to establish confidence in His command. Even if the speech is with someone else, it refers to all". Al-Qushayrī (2000) emphasises that the true story means "the dust of confusion has no authority over the witnessing of divine unity. No delusion of any created being reaches the secret of His *ḥukm* (ruling). Nothing that is known in existence can come close to it, and nothing that is imagined can conceive of the *taqdīr* (divine decree)".

It is worth noting that the next phrase "there is no god but God" is translated from the Arabic "*mā min ilāhin illāLlāh*", which differs from the usual *shahādah* or Islamic testimony of faith, "*lā ilāha illāLlāh*". The latter is the standard Islamic testimony of faith *lā ilāha illāLlāh* that appears twice in Q37:35 and Q47:19. In addition, *lā ilāha illā Hū* (there is no god, but He) also appears in Q2:163, Q2:255, Q3:2, Q3:6, Q4:87, Q6:102, Q7:158, and in many other places (Nasr et al. 2017). Within the text of the Qurʾān, the *shahādah* takes the form *there is no god, but He* much more often than *there is no god, but God*. Nevertheless, some scholars, such as Al-Rāzī (1981) and Al-Zamakhsharī (2009) (d. 538/1144), interpret this "*mā min ilāhin illāLlāh*" deviation as a means of intensifying the negation of other gods. As an essence of Islamic theological framework encapsulated in the opening statement of the *shahādah*, the phrase affirms the indivisible unity of the divine and the exclusive veneration due to this "Supra-Being". Likewise, at the core of Qurʾānic doctrine lies the affirmation of monotheism, known as *tawḥīd*, while its antithesis, *shirk* (polytheism), bears the weight of critical scrutiny. The unequivocal and resolute repudiation of *shirk* in the Qurʾān is a testament to the unwavering commitment to monotheism. That is how the Qurʾān comes to emphasise and re-emphasise the oneness and majesty of God. Narrative, truth, and the testimony of faith are intimately linked in the single verse of the Qurʾān.

*4.2. Narrative and Beauty*

Verse 3 of *Sūrah Yūsuf* is possibly the most significant corpus that uses the term *qaṣaṣ* in describing the *phenomenon* and *noumenon* of Qurʾānic narrative.

> We recount unto thee the most beautiful of stories by Our having revealed unto thee this Qurʾān, though before it thou wert among the heedless. (12:3)

Contemplating the interpretations of Muslim exegetes on the rationale behind the *Sūrah Yūsuf*, as well as Prophet Joseph's stories being deemed the "most beautiful of stories", may aid in comprehending this notion and illuminating the essence of a "narrative." This verse implies that the term *qaṣaṣ*, in its Qurʾānic sound, does not only relate to sayings regarding the Qurʾān, religion, or even pre-Islamic prophets but that *qaṣaṣ* may be a part of the Qurʾān in some form. The identification of God's revelation to the Prophet as *qaṣaṣ* here is particularly interesting, as *Sūrah Yūsuf*, at the beginning of which these verses appear, is a long narrative section about a pre-Islamic prophet. Clearly, the Qurʾān is described as the most beautiful discourse, part of which is the tale of Joseph, which is referred to as the most beautiful of stories here,

These are the signs of the clear Book. (2) Truly We sent it down as an Arabic Qurʾān, that haply you may understand. (3) We recount unto thee the most beautiful of stories by Our having revealed unto thee this Qurʾān, though before it thou wert among the heedless.

In relation to the understanding of *aḥsan al-qaṣaṣ* as "the most beautiful of stories,", al-Qurṭubī (d. 671/1273) alludes the story is characterised in this way because it is a profound amalgamation of both the realities of one's religious life as well as one's worldly truths (Al-Qurṭubī 2003). Often there is a deeper motive behind these hair-splitting semiotics (Rasdi et al. 2021). According to Al-Qurṭubī (2003) and Ibn ʿAjībah (2002), the narrative speaks about the divine unity, and the essence of celestial beings such as prophets, angels, and saints, as well as the nature of individuals who possess knowledge or ignorance. Additionally, it delves into the intricacies of dream interpretation, forgiveness, and the boundless nature of divine love. Simultaneously, it also sheds light on the complexities of earthly matters such as governance, tribulations, retribution, human relationships, commerce, social norms, politics, and the art of deception.

Fittingly, Ibn Taymiyyah (1995) restricts *aḥsan al-qaṣaṣ* as narrativisation because in this verse, God does not claim to narrate the most beautiful "narratives" (*qiṣāṣ*) to human beings, but rather the best "narrativisation" (*qaṣaṣ*). Al-Rāzī (1981) expounds that the essence of a story lies in its gradual revelation of events through narration, hence earning the title "narrative". In this case, he elucidates that the term "most beautiful" (*aḥsan*, derived from the root *ḥ-s-n*, meaning beauty) pertains to the *ḥusn al-bayān* (artistry of expression) rather than the narrative content. As expounded, the essence of the concept of beauty lies in the unparalleled mastery of language, where vocabulary is so eloquent that it becomes impossible to imitate. He posits that the narrative of the Qurʾān in question is chronicled in the historical literature, yet none of these accounts parallel *sūrah*'s *faṣaḥah* (clarity) and *balāghah* (eloquence). Al-Rāzī further explains that *aḥsan* encompasses not only the narrative itself but also the profound insights, ethical principles, and mystical revelations it imparts. This treasure of wisdom cannot be gleaned from any other source, rendering the story a unique and invaluable repository of knowledge. By doing so, he alludes to the reader's own experience of pleasure, whether it is merely aesthetic on the level of language from the stylistic beauty or intellectual pleasure gained on the level of meaning, indicating moral fulfilment from the tales.

Furthermore, regarding the story of Prophet Joseph in the *Sūrah Yūsuf* particularly, Al-Thaʿlabī (2002) expounds on the multifarious rationales behind this phenomenon. In the words of al-Thaʿlabī, the story of Prophet Joseph is the most beautiful "because the lesson concealed in it, on account of Yusuf's generosity and its wealth of matter, in which prophets, angels, devils, jinn, men, animals, birds, rulers, and subjects play a part". He perceives this tale as the epitome of storytelling due to its prolonged duration. Per their account, there is a general agreement that the time span between Joseph's initial dream as a juvenile and his eventual reunion with his kinfolk amounts to four decades. Cited in *Maʿālim al-Tanzīl* (Al-Baghawī 1989), Ibn ʿAṭā Allāh al-Iskandarī (d. 709/1310) emphasises that the story of Joseph holds within it a profound therapeutic power, thus making it a prime example of what constitutes "the most beautiful of stories". The contention put forth is that the narrative of Joseph has a calming effect on any individual who is experiencing distress.

The renowned Sufi exegete, Al-Qushayrī (2000) goes on to elaborate that the beauty of this narrative lies in its lack of *al-amr wa l-nahy* (commanding and forbidding), which often triggers feelings of inadequacy and limitation. By transcending these limitations, the narrative becomes a symbol of the limitless potential of the human spirit. Al-Qushayrī (2000) posits that the most profound narrative is not overtly didactic (command or forbid) yet still imparts ethical values and virtuous conduct that is both uplifting and potentially emulated. The augmentation of the aesthetic quality of action can be perceived in two distinct manners: either the intrinsic moral worth in spiritual parlance is amplified, thereby leading to an augmented recompense, or the apprehension of the act of performing good is progressively intensified in its own right as the person persists in it (moral gratification

and felicity). The essence of beauty lies in the act itself, and its allure is amplified as it becomes ingrained in the being of the doer until it becomes an effortless expression of their being. A high-calibre narrative inevitably references 'the beloved' in the context of al-Qushayrī's vocation and elucidation; this allusion may pertain to the divine, prophets, and messengers as depicted in the tale of Joseph. In a tale of human nature, it may allude to characters who are deemed 'admirable' or 'virtuous.' Ultimately the designation of "the most beautiful of stories" shall endure as a divine construct, as suggested by al-Qushayrī. All other narratives, however, shall inevitably bear the mark of imperfection.

### 4.3. Narrative and Explication

The dictum of a narrative is marked by the refinement and sensitivity that lend the narrator their cultured quality, which is dependent not only on wit and intelligence but on the profundity of their emancipation as a measurement of these qualities. Building on the concept of narrative pedagogy, I have clustered four forms of *qaṣaṣ* in the domain of narrative and explication. These forms, (1) Q7:176 *faqṣuṣ al-qaṣaṣ* (recount the stories), (2) Q12:111 *fi qaṣaṣihim ʾibrah* (in their stories is a lesson), (3) Q18:64 *fartaddā ʿalā ʾāthārihimā qaṣaṣā* (So they turned back, retracing their steps), and (4) Q28:25 *waqaṣṣ ʿalayh al-qaṣaṣ* (and recounted his story unto him), are used to convey a range of meanings, including narration, mention, declaration, explanation, recountment, explication, clarification, and tracking. In his dictionary of the Qurʾānic lexicon, *al-Mufradāt fī Gharīb al-Qurʾān*, al-Rāghib Al-Iṣfahānī (2009) (d. 502/1109) elucidates the form *qaṣaṣ*, saying "*al-qaṣṣu is tatabbuʿ al-āthār* (that is to follow the traces or footsteps)." The definition of the word *qaṣaṣ* in Arabic is derived from the verb *qaṣṣ*, which means to cut, to clip, to cut off, to shear, to curtail, to scissors, chips, cuttings, to match, to retaliate, reprisal, to follow up, to track, tracker, or breastbone (Badawi and Muhammad 2008). It is used to refer to the act of following someone's footsteps. To follow their footsteps at night is called "*taqṣaṣ*." Some people use the word *qaṣaṣ* to mean simply following someone's trail, while others use it specifically to refer to telling stories. One of the precise meanings of "*qaṣaṣ*" is to follow in someone's footsteps at any time, as in Q18:64. The word describes two people who turned back to follow in their footsteps (Al-Ṭabarī 2001).

In the short haul, to follow someone's footsteps of *qaṣaṣ* symbolises a view the pedagogue held previously, and the student followed later. It is a well-known fact that the etymology of the term pedagogy traces back to the Greek word *paidagōgos*, "slave who escorts boys to school and generally supervises them," later "a teacher or trainer of boys," from *pais* (genitive *paidos*) "child" and *agōgos* "leader," from *agein* "to lead" (Harper n.d.b). This amalgamation of words gives rise to the concept of conveying the narrative and communicating between the pedagogue and learner. "Narrative" is taken in its common sense of a recounted story, and I make specific appeal to the characteristics of narrative and explication with a view to establishing a pedagogical purpose for narrative in Islamic education. Explication is taken in the interpersonal sense of the term, rather than in the broader inferential sense. The concept of explication involves the act of allocating concepts and is offered neither descriptively nor stipulative as prominently described by Rudolf Carnap (1950). This process of explanation, especially of the meaning of a sentence or passage, literally "an unfolding," can be traced back to its origins from French *explication*, from Latin *explicationem* (nominative *explicatio*), noun of action from past-participle stem of *explicare* "unfold; explain," from ex "out", *plicare* "to fold" (from PIE root *plek- "to plait") (Harper n.d.a). Perhaps it is through this act appropriately referred to as "explication" that a profound and meaningful connection with the text is established. Likewise, we have come to the elucidation between explication, narrative or *qaṣaṣ*, and the two previous domains, namely, truth and beauty. The below-cited verses in Q7:176, Q12:111, Q18:64, and Q28:25 cohere with a broader Qurʾānic tracts concerning the narrativisation, reflection, lesson for intellectuals,

(i) So, recount the stories, that haply they may reflect. (7: 176)

This verse should be read in the context of the preceding verse 175 of *Sūrah al-Aʿraf*, which mentions people who rejected their prophets and the messages. A central point in this narrative explication is the general warning about the moral fate of those who "cast off" and deny the signs that God has sent them. Although Al-Rāzī (1981) favours this broad interpretation, many of the earliest exegetes, like Al-Ṭabarī (2001) and Al-Zamakhsharī (2009) (d. 538/1144) took *ʾalladhī ātaynāhu ʾāyātinā* (the one to whom We gave Our signs) to be Balʿām Bāʿūrāʿ (Balaam of Beor), a man who was spiritually endowed and, therefore, able to receive some Divine messages and have his supplications answered by God. Both passages, according to al-Rāzī, warn everyone to learn that the spiritual knowledge God has given them may be taken away if they lean towards base desires and earthly problems. He bases this on the *ḥadīth*, "Whoever increases in knowledge, but does not increase in guidance, increases in nought but the distance from God." The Prophet is instructed to tell the stories and the accounts of former communities that are mentioned in this *Sūrah that* are meant to be told (Al-Ṭabarī 2001; Al-Rāzī 1981). A point should be stressed here. When it comes to recounting the narrative, this verse employed *fa-qṣuṣ al-qaṣaṣ* (to mention, recount, relate, narrate, track, tell, declare, clarify, explain) (Badawi and Muhammad 2008) because it accommodates a notion of strong association with bodily posture, direct speech, voice, and performative and narrative character.

(ii)   Certainly, in their stories is a lesson for those possessed of intellect. It is not a fabricated account; rather, it is a confirmation of that which came before it, and an elaboration of all things, and a guidance and a mercy for a people who believe (12: 111).

And in a similar vein, just as the "prologue" of *Sūrah Yūsuf* mentioned earlier introduces the Qurʾān's unique style of narrativisation as well as most beautiful stories, the "epilogue" of the *Sūrah Yūsuf*, particularly verse 111, *fi qaṣaṣihim ʿibrah* (in their stories is a lesson), provides a profound insight into the essential elements of narrative and elucidates why its methodology is most suited to fulfil the Qurʾān's ultimate purpose of providing "guidance and mercy for those who believe." The term *qaṣaṣihim* (their stories) refers to the accounts of the prophets cited in the Qurʾānic text (al-Maḥallī and al-Suyūṭī 2007). It is important to note that the Qurʾān serves to affirm the validity of the preceding scriptures, as evidenced in verse 37 of *Sūrah Yūnus* (This Qurʾān could not have been fabricated [by anyone] apart from God; rather, it is a confirmation of that which came before it, and an elaboration of the Book in which there is no doubt, from the Lord of the Worlds).

What can be better assessed, in particular, is what verse 111 of *Sūrah Yūsuf* refers to *qaṣaṣ* as *ʿibrah* or *a lesson for those possessed of intellect*. Amidst the narrative of the Qurʾān, Ibn Jamāʿah handed down to us seven profound insights regarding the recurring lessons. Although the existence of his *al-Muqtanaṣ fī Fawāʾid Tikrār al-Qaṣaṣ* cannot be ascertained, Al-Zarkashī (2006) (d. 794/1392) and Al-Suyūṭī (1996) (d. 911/1505) have meticulously documented these seven lessons in their works, *al-Burhān* and *al-Itqān*, respectively: (1) as we journey through the tales, we uncover deeper layers of meaning, gradually unravelling their complexities with each passing moment; (2) it is imperative that individuals, regardless of their level of religiosity or intellectual prowess, possess a comprehensive understanding of the fundamental narratives presented in the Qurʾān; (3) the Qurʾān showcases its unparalleled eloquence by presenting a single scene through a multitude of compatible portrayals, thereby revealing the depth and complexity of its meaning; (4) the dissemination of knowledge is best achieved through the art of storytelling, as human beings are inherently driven to share narratives rather than mere directives; (5) the inability of the masters of Arabic rhetoric to match even a single variant of the Qurʾānic narratives serves as a humbling reminder of the limitations of human knowledge and understanding; (6) anticipating the argument of polemicists who assert that the Qurʾān lacks the ability to present its narratives in diverse manners, one may question why those who reject it cannot undertake a similar task; and (7) the pursuit of captivating the minds of readers, who seek to submerge themselves in these scenarios repeatedly, is achieved by presenting an exceptionally novel portrayal on every occasion (Al-Suyūṭī 1996).

The list of Ibn Jamāʿah's insights for the Qurʾān depicting narrative could be considered comprehensive if it were not for another purpose of narrative pedagogy, namely, to keep the earlier prophets', or saints', "memory alive". Also, this narrativising of Qurʾānic stories as providing a lesson or bridge to oneself is comparable to Ibn al-ʿArabī's view of man as advancing to man as a "spiritual being" (Clark 2001), insofar as both authors try to propose ways of finding meaning through the Qurʾān. For all intents and purposes, this great mystic and thinker, Muhyīddīn Ibn al-ʿArabī (d. 638/1240), alludes the interconnection of ʿibrah (a lesson, crossing over to the other side),

> Reporting (*ikhbār*) about things is called "expression" (*ibāra*) and interpreting dreams is called "interpretation" (*taʿbir*). This is because the expresser/interpreter "crosses over" (*ʿubur*) by means of what he says. In other words, by means of his words he passes (*jawāz*) from the presence (*ḥaḍra*) of his own self to the self of the listener. Hence, he transfers his words from imagination to imagination, since the listener imagines to the extent of his understanding. Imagination may or may not coincide (*taṭābuq*) with imagination, that is, the imagination of the speaker with that of the listener. If it coincides, this is called his "understanding" (*fahm*); if it does not coincide, he has not understood . . . We only make this allusion to call attention to the tremendous of imagination's level, for it is the Absolute Ruler (*al-ḥākim al-muṭlāq*) over known things. (Chittick 1989)

(iii) He said, "That is what we were seeking!" So they turned back, retracing their steps. (18: 64)

This verse 64 of *Sūrah al-Kahf* is preceded and succeeded (Q18:60-82) by the account of Moses and the mysterious "servant" he meets at *majmaʿ al-baḥrayn* (the junction of the two seas). Generally, in this verse, we find *fartaddā ʿalā āthārihimā qaṣaṣā* (So they turned back, retracing their steps) in reference to a story of Prophet Moses and his servant, widely identified as Yūshaʿ ibn Nūn (Joshua) retracing their earlier footsteps (Al-Ṭabarī 2001; Al-Rāzī 1981; Al-Qurṭubī 2003). Scholars have observed that the servant, similar to Moses, was aware of their quest to locate the confluence of the two seas. Some accounts suggest that he possessed knowledge that the loss of their fish would serve as an indication of their arrival at the destination (al-Maḥallī and al-Suyūṭī 2007). However, despite witnessing the fish's disappearance, the servant succumbed to slumber and failed to recollect the matter upon awakening. The servant endeavoured to justify his lapse by attributing it to Satan's influence, citing that he had failed to communicate the issue of the fish to Moses (Al-Qurṭubī 2003). As the servant narrates the fish incident to Moses, he expresses his astonishment towards it. Upon discovering the fate of the fish, Moses exclaimed, *dhālika mā kunnā nabghī* (That is what we were seeking!). It became clear to him, or as some sources suggest, he was informed beforehand, that the location where the fish disappeared into the sea was the exact spot where he would encounter a servant of God possessing greater knowledge than himself (Q18:60). Therefore, Moses was cognisant of the fact that it was imperative to revisit that location. Consequently, they reversed their course by retracing their previous path, *fartaddā ʿalā āthārihimā qaṣaṣā* (So they turned back, retracing their steps) (Ibn Kathīr 1997). When interpreted as an allegory or a symbolic narrative of the soul's journey, Moses retracing his path signifies a spiritual return to God, undoing the "fall" into worldly existence and detachment from God. In his allusive exegesis, Al-Kāshānī (2021) states, "*fartaddā ʿalā āthārihimā qaṣaṣā* (So they turned back, retracing their steps) in rising to the station of the first *fiṭrah ūlā* (primordial nature) as they had initially been retracing their footsteps at the descent from the ascent to perfection until they found the holy intellect, which is one of God's servants singled out for the privilege of [divine] solicitude and mercy."

(iv) When he came and recounted his story unto him, he said, "Fear not. You have been saved from the wrongdoing people." (28: 25)

Most exegetes consider the name of the *Sūrah al-Qaṣaṣ* name of Surah 28, derived from the narrative in which Moses recounted his escape from Egypt to his future father-in-law. In

verse 25 of *Sūrah al-Qaṣaṣ*, referred to above, Al-Ṭabarī (2001) mentioned the story in which Moses went to the father of the women and *waqaṣṣ ʿalayh al-qaṣaṣ* (and recounted his story unto him), of his flight from Egypt, and the father told him to *fear not*, because Pharaoh had no power in that land. Most Muslims say that the father of the women, named Jethro in the Bible, was identical to the prophet Shuʿayb, though the Qurʾān does not explicitly indicate this (Al-Rāzī 1981).

## 5. Discussions

Regardless of the various possibilities pertaining to andragogy, heutagogy, critical pedagogy, dialogic pedagogy, etc., of the contemporary framework, the teaching and learning of Qurʾān traditionally presupposed a venerable pedagogy of "master-disciple" rapport involving essential prerequisites of needs, special qualities of vision, and many others (Morris 2003). More importantly, the Qurʾān (7:176) imperatively states *fa-qṣuṣ al-qaṣaṣ* (so recount the stories) as the explication of *taqṣīṣ* in the Islamic didactic ethos presupposes the concept of knowledge as "fixed, memorable truth." Internalising this account, one can assume that the narrativity of the Qurʾān was performed with the aim of attaining both significant visual piety and oral piety. The two modes of learning, namely "learning by explicating" and "learning by listening," symbolise the relationship stemming from *ādāb al-ṣuḥbah* (etiquettes of companionship), which provides a narrative pedagogy model of teaching and learning between the pedagogue and learner. Michael Sells (2007) also suggested another feature we must draw attention to, if only briefly and inadequately,

> The experience of the Qurʾān in traditional Islamic countries is very different from Western attempts to read it as a story bound within the pages of a book with a sequence of beginning, middle, and end. For Muslims, the Qurʾān is first experienced in Arabic, even by those who are not native speakers of Arabic. In Qurʾān schools, children memorize verses, then entire Suras. They begin with the Suras that are at the end of the Qurʾān in its written form. These first revelations to Muhammad express vital existential themes in a language of great lyricism and beauty. As the students learn these Suras, they are not simply learning something by rote, but rather interiorizing the inner rhythms, sound patterns, and textual dynamics—taking it to heart in the deepest manner.

Whether deemed classic or contemporary, the manifestation of narrative pedagogy does not arise from a place of pure intellect or a secluded vacuum but rather exists within the confines of history, language, and societal context. The impact of the Qurʾān on narrative has been a topic of discussion, as it is believed to hold significant sway over society's educational and cultural norms. The subject matter at hand, which engages God as the speaker, presents a formidable challenge regarding its measurement. The Qurʾān elucidates that divine text is disclosed in the tongue of the individuals it is designated for. Consequently, it encompasses constituents that reverberate with the storyline anticipations of its proposed recipients. The audience, deeply immersed in Qurʾānic verses and the tradition of the narrative literature, employed it as a tool to mould and perfect their principles, past, and sense of self. In conveying narrative, then, Qurʾānic stories should be made central and other narratives of Islamic biography or history taught in relation to it. Perhaps through typological figuration, the contemplation of the listener leads to a re-evaluation of conventional notions surrounding the dynamics between a student and teacher, as well as the dissemination of knowledge within a pedagogical setting. The Qurʾān tells stories about prophets and their communities, such as Adam, Noah, Abraham, Moses, Jesus and many others. These stories are not just narratives but also representations of how people should live according to God's commandments. Through these stories, the Qurʾān invites its readers to reflect on their own lives and actions. By doing so, they become part of the story themselves. They become actors who are called upon to live out the values and principles that are embodied in the stories. As described by Lawson (2022), in this way, *diegesis* becomes *mimesis* because the stories are not just being told; they are being lived out by those who read them.

As mentioned earlier, the term "explication" is employed in an interpersonal context, rather than in a broader inferential sense. From this perspective, I propose that Muslims should perceive this shift as a transition towards a spiritually oriented intellection (Al-Attas [1995] 2014) and conceptualisation of the ontic-ontological significance of narrative pedagogy in Islamic education. This perspective sees the individual as consistently shaped through an ongoing process of narrative explication, which occurs within a pedagogical relationship that is deeply intertwined with models of sense-making mediated by self-religiosity. Narration, serving as a communicative medium, fulfils at least a dual role: elucidating textual meaning and retracing historical events. Given the Qurʾān's guided role in revitalising and reshaping Muslims, this transition necessitates the recognition of the reciter or educator as playing a fundamental role in how listeners interpret their identities and positions within the world. Through an emphasis on interpreting and imagining as shared endeavours, interpretive pedagogies like narrative pedagogy, encourage both educators and learners to collectively harness their insights. This approach prompts them to question their preconceptions, imagine novel avenues for nurturing spirituality, and actively collaborate with others in order to promote theological reflection, spiritual growth, and social empowerment.

Hanna Meretoja (2018) argues that the process of self-reflection and self-transformation through narratives occurs when individuals engage with literary and various cultural narratives. Our lives are contemplated in connection with the narratives we encounter through hearing and recitation. Hans-Georg Gadamer ([1960] 1997) underscores the notion that our self-awareness is profoundly shaped by engaging with others, especially those with differing perspectives. This recognition of the other's standpoint becomes instrumental in revealing our inherent preconceptions, constituting a pivotal aspect of how narrative and various forms of pedagogy facilitate the process of self-examination and self-awareness. This point is in alignment with Abdallah Rothman's (2022) recent Islamic model of the soul, where the research interest lay in developing a foundational model for Islamic approaches to psychotherapy, as well as for an Islamic theory of human psychology. When al-Ghazālī advises us that "the soul of human being is as a mirror," he envisions his audience to find "the Real" therein (Zargar 2017). Shifting our focus in this manner enables us to recognise that the inquiry into the connection between narrative and the soul inherently encompasses the subject's interaction with social practices and the intricate power dynamics at play.

In a broader context, the proposed framework (truth, beauty, and explication) requires practical implementation in the realm of Islamic education, as well as studies that observe key concepts related to narrative, experience, subjectivity, memory, and their interrelations. The framework holds potential value for educators, policymakers, and educational authorities, particularly in Islamic countries, by highlighting the significance of narrative pedagogy in fostering theology, spirituality, and mimetic function skills among students. In the future, it may serve as a catalyst for generating innovative ideas, fresh teaching materials, and innovative pedagogical modules. Additionally, conducting ontic-ontological studies of the Qurʾānic concept would enable researchers to explore broader questions of pedagogy within Qurʾānic studies and Islamic education. This entails not only *tadabbur* (contemplation) of the narratives within the Qurʾān but also a comparative analysis of interpretative patterns and the human soul between the Qurʾānic narrative and other narrative traditions where the Qurʾān is not the central focus. How can studies of narrative contribute to cross-disciplinary dialogues within Islamic education regarding the interpretation of Qurʾānic stories, exegeses, and foundational theosophical and philosophical texts? What guidance might the framework offer educators in other, more practical fields of Islamic education, including the rapidly growing domain of digital Islamic studies? Alternatively, what insights and tracts from this presentation can educators in Islamic education glean?

## 6. Conclusions

From the outset, it has been my intention to establish the fundamental conceptual tracts of narrative in the Quran, i.e., *qaṣaṣ*. Through an exploration of the Qurʾān's intri-

cate conceptual networks, I aimed to demonstrate how these tracts are established and how they can be observed in works of Qurānic exegeses and Islamic intellectual tradition. By saying this, to successfully construct a religiously suitable theory that includes major features from the Islamic tradition, it is critical to establish the framework for an Islamic paradigm of narrative pedagogy with indirect input from Muslim luminaries, Islamic scientist philosophers, Islamic scholars, and source materials. At this juncture, it is through deep contemplation and reflection that one can gain a deeper understanding and appreciation of the rich theosophical and philosophical insights contained within these texts. The essence of our discourse was to discern a term that could aptly capture the concept of narrative pedagogy in the context of Islamic education. One may posit that the term *qaṣaṣ* in the form of a noun, as found in the Qurān six times, bears the potential to signify the pinnacle narrative pedagogy, as its semantic breath conveys the notion of "the narration, declaration, clarification, explication and tracking" along with the purposes, functions and interpretations of those verses.

Furthermore, the concept of narrativisation, also known as *taqṣīṣ*, is widely acknowledged as the predominant explication. In the pursuit of understanding the *qaṣaṣ*, it is imperative to acknowledge the various appellations it is referred to, including *nabā*, *ḥadīth*, *siyar*, and the like. However, this inquiry aims to illuminate the significance that sets *qaṣaṣ* apart from the rest and certainly creates gaps, showing that further research is needed for other terms. Ultimately, I invite the reader to ponder deeply on the subject of the narrative, taking into account the normative and elaborative on the truth, beauty and explication of the term *qaṣaṣ*. The triad of truth, beauty and explication are fundamental pillars within the Islamic framework for narrative pedagogy, representing the essence of the human condition in relation to education. Because these domains emerge from the concept of *qaṣaṣ*, I conclude by suggesting that narrative has been integral to Islamic education at many points in its history, so that narrative approaches are more traditional than they might seem, as can be treasured in light of the verse of the Qurān, *fa-qṣuṣ al-qaṣaṣ*.

**Funding:** This research was funded by the Ministry of Higher Education (MOHE) of Malaysia under the Fundamental Research Grant Scheme (FRGS/1/2021/SSI0/USIM/02/5).

**Conflicts of Interest:** The author declares no conflict of interest.

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
