# Peer review of "Tracing the Tracts of Qaṣaṣ: Towards a Theory of Narrative Pedagogy in Islamic Education"

_religions, doi:10.3390/rel14101299_

Round 1

Author Response

Dear reviewer,

Thank you for providing me with the opportunity to enhance the quality of my submitted manuscript. Below, I have summarised the changes made in response to the reviewer's comments in a point-by-point manner.

Comment 1: It would be appropriate to explain why the author opts for the approach "narrative pedagogy" and how distinguishes these two approaches "narrative pedagogy" and "narrative method".

Response 1: Thank you for your advice. In Section 3.1, I concentrated on exploring narrative as pedagogy and its role within other discourses. I have highlighted and added some sentences to differentiate these approaches.

(Line 158-160) The act of introspectively contemplating and analysing the stories that shape one’s existence during the educational journey defines the concept of narrative pedagogy.

(Line 203-208) In due course, Paul Ricoeur delved deeper into these aspects, dedicating his later endeavours to the contemplation of narrative and its significance in the realm of philosophy, including the philosophy of history and the metaphysical implications of time (Joy 1997). As Ricoeur (1991) puts it, hermeneutics represents a philosophy deeply rooted in the concept of perpetual mediatedness. It underscores that no form of self-understanding exists devoid of mediation through signs, symbols, and texts.

(Line 223-231) Clearly, Hinchman and Hinchman’s definition and characterisation of narratives stress the significance of social interaction in constructing narratives and transforming human experience into meaning. According to Rogers (2011), narrative pedagogy is “any educational method that intentionally uses the narrative arts of storytelling, drama, and creative writing to nurture theological reflection, spiritual growth, and social empowerment.” Hence narrative pedagogy should be discerned from narrative method, as the former is deliberate strategy by teacher or educator to nurture theology and spirituality which has a mimetic function, whereas the latter constitutes temporality; voice (who speaks?); vision or focalisation (who perceives?); and style (James and Wayne 2007).

Comment 2: Content of subchapter 3.2. “Narrative Pedagogy in the Islamic Intellectual Tradition: A Selection of Approaches” is not clearly and systematically argumented. The selected approaches are written descriptively without providing critical review.

Response 2:  In the explanation of the content in subchapter 3.2, I started with an overview of narrativity in the Islamic intellectual tradition, which manifested in a specific role in Muslim society, i.e., quṣṣāṣ (storytellers). Through a literature review, I categorized the samples into two major schools of thought involved in the transmission and communication of religion: theologians and philosophers. The specific approach results are presented in the samples of al-Ghazali and Ibn Sina.

Comment 3: Title 3.3. The Qurʾān and Narrative Literature is not sufficiently precise and clear in relation to the content.

Response 3: Thank you for your suggestion. In the second and third paragraphs of Section 3.3, I have provided a detailed elaboration of this content.

(Line 340-353) The intricate relationship between the Qurʾān and subsequent literary works is inherently nuanced and multifarious. The Qurʾān possesses the potential to not only mould literary dialogue straightforwardly but also serves as an autonomous frame of reference to rejuvenate other notions and musings. Much like other religious and literary traditions, Islamic literature showcases various customs that can be categorised into three types - those derived from the holy book through direct and rational means, those that align with its essence, and those completely unrelated to it (Bin Muhammad Yusoff and Mohd Yusuf 2023).

The narratives found within the Qurʾān possess a distinct historical context and are intricately linked three specific categories or temporal phases in history: the past, the present, and the future. Bennabi’s (2004) insights effectively encapsulate this perspective regarding narratives within the Qurʾān. Focusing on narrative literature, Roberto Tottoli (2017) retraces literary legacy and how it has been influenced, or in other words, the mark left by the Qurʾānic narratives on Islamic Arabic literature.

Comment 4: In the discussion and conclusions, it is necessary to explain more broadly and thoroughly the intentions and potentials of applying the narrative method in Islamic education. Less emphasis is placed on this issue in the paper.

Response 4: Thank you for this important comment. In subchapter 5. Discussion, I have included additional discussion as follows:

(Line 846-897) As mentioned earlier, the term “explication” is employed in an interpersonal context, rather than in a broader inferential sense. From this perspective, I propose that Muslims should perceive this shift as a transition towards a spiritually oriented intellection (Al-Attas 2014) and conceptualisation of the ontic-ontological significance of narrative pedagogy in Islamic education. This perspective sees the individual as consistently shaped through an ongoing process of narrative explication, which occurs within a pedagogical relationship that is deeply intertwined with models of sense-making mediated by self-religiosity. Narration, serving as a communicative medium, fulfils at least a dual role: elucidating textual meaning and retracing historical events. Given the Qurʾān’s guided role in revitalising and reshaping Muslims, this transition necessitates the recognition of the reciter or educator as playing a fundamental role in how listeners interpret their identities and positions within the world. Through an emphasis on interpreting and imagining as shared endeavours, interpretive pedagogies like narrative pedagogy, encourage both educators and learners to collectively harness their insights. This approach prompts them to question their preconceptions, imagine novel avenues for nurturing spirituality, and actively collaborate with others in order to promote theological reflection, spiritual growth, and social empowerment.

Hanna Meretoja (2018) argues that the process of self-reflection and self-transformation through narratives occurs when individuals engage with literary and various cultural narratives. Our lives are contemplated in connection with the narratives we encounter through hearing and recitation. Hans-Georg Gadamer (1997) underscores the notion that our self-awareness is profoundly shaped by engaging with others, especially those with differing perspectives. This recognition of the other’s standpoint becomes instrumental in revealing our inherent preconceptions, constituting a pivotal aspect of how narrative and various forms of pedagogy facilitate the process of self-examination and self-awareness. This point is in alignment with Abdallah Rothman’s (2022) recent Islamic model of the soul, where the research interest lies in developing a foundational model for Islamic approaches to psychotherapy, as well as for an Islamic theory of human psychology. When al-Ghazālī advises us that “the soul of human being is as a mirror,” he envisions his audience to find “the Real” therein (Zargar 2017). Shifting our focus in this manner enables us to recognise that the inquiry into the connection between narrative and the soul inherently encompasses the subject’s interaction with social practices and the intricate power dynamics at play.

In a broader context, the proposed framework (truth, beauty, explication) requires practical implementation in the realm of Islamic education, as well as studies that observe key concepts related to narrative, experience, subjectivity, memory, and their interrelations. The framework holds potential value for educators, policymakers, and educational authorities, particularly in Islamic countries, by highlighting the significance of narrative pedagogy in fostering theology, spirituality, and mimetic function skills among students. In the future, it may serve as a catalyst for generating innovative ideas, fresh teaching materials, and innovative pedagogical modules. Additionally, conducting ontic-ontological studies of the Qurʾānic concept would enable researchers to explore broader questions of pedagogy within Qurʾānic studies and Islamic education. This entails not only tadabbur (contemplation) of the narratives within the Qurʾān but also a comparative analysis of interpretative patterns and human soul between Qurʾānic narrative and other narrative traditions where the Qurʾān is not the central focus. How can studies of narrative contribute to cross-disciplinary dialogues within Islamic education regarding the interpretation of Qurʾānic stories, exegeses, and foundational theosophical and philosophical texts? What guidance might the framework offer educators in other, more practical fields of Islamic education, including the rapidly growing domain of digital Islamic studies? Alternatively, what insights and tracts from this presentation can educators in Islamic education glean?

Reviewer 2 Report

Overall, the author does a good job of laying out his argument about what narrative is in Islamic pedagogy and how that narrative is used to convey various kinds of knowledge. The author did a particularly good job of addressing the layered-ness and interconnection among and within Islamic narrative sources. The author does a well in grounding their discussion of narrative and pedagogy in classical Islam with references and comparisons to narrative as a teaching tool within the Western and, especially, the Classical tradition.

The author seems to have a good grasp of the current, relevant literature relating to the topic.

I do wish that the author explained earlier in the article what qasas is. I realize that the full explanation is very long and, hence, was delayed to later in the article, but a brief gloss or definition nearer to the beginning of the article would be helpful for readers coming from outside Islamic studies.

Some minor quibbles:

Page 1 Line 28: I think I understand what is being said here, but do not find the meaning of the “bound in chains” metaphor all that clear. Is there a way that the author could make their characterization of contemporary knowledge practices?

Page 1 Line 43: “Any other” what?

Page 2 Line 78: I am not sure that “redress” is the word the author is looking for? What is the wrong that is being put right here?

Page 2 Line 82: Would “indeed” be better than “however” here? As the sentence beginning with “however” and the sentence directly preceding it both talk about the use of constructivism in Islam, I do not see the contrast signaled by the “however.”

Page 13 Line 629: Should it read “4 forms of qasas”?

The English in the article is standard academic English. There are no substantial problems with the English that cannot be addressed with light copy editing. 

Author Response

Dear reviewer,

Thank you for providing me with the opportunity to enhance the quality of my submitted manuscript. Below, I have summarised the changes made in response to the reviewer's comments in a point-by-point manner.

Comment 1: I do wish that the author explained earlier in the article what qasas is. I realize that the full explanation is very long and, hence, was delayed to later in the article, but a brief gloss or definition nearer to the beginning of the article would be helpful for readers coming from outside Islamic studies.

Response 1: Thanks for your advice. Under subchapter 2. Materials and Methods, I've added the following sentence (Line 133-135)
When it comes to narration, the Qurʾān often uses the verb form qaṣṣa to relate stories and histories, while the noun form qaṣaṣ is used to denote an account or report (Badawi and Muhammad 2008).

Comment 2: Page 1 Line 28: I think I understand what is being said here, but do not find the meaning of the “bound in chains” metaphor all that clear. Is there a way that the author could make their characterization of contemporary knowledge practices?

Response 2: (Line 26-28) Amidst the multitude of signifiers that generate diverse contexts, contemporary knowledge practises appear to be bound by the chains of overstimulation, particularly through technology. 

Comment 3: Page 1 Line 43: “Any other” what?

Response 3: (Line 42-43) He asserts that narrative connects Sufi and philosophical virtue ethics better than any other common threads. 

Comment 4: Page 2 Line 78: I am not sure that “redress” is the word the author is looking for? What is the wrong that is being put right here

Response 4: (Line 78-82) Adjacent to the backdrop of these intellectual traditions, there is a need for a theoretical framework that allows Muslims to explore the purposeful complexity of the roles that narratives play in their lives. In short, I endeavour to bring to the forefront the narrative pedagogy of Islamic intellectual tradition by redressing the tracts, purposes, or maqāṣid (objectives) of the Qurʾān. 

Author Response

Dear reviewer,

Thank you for providing me with the opportunity to enhance the quality of my submitted manuscript. Below, I have summarised the changes made in response to the reviewer's comments in a point-by-point manner.

Comment 1: In these lines (and not only here) you are very clear in your purpose. However, please shed some more light on the reasons why. Why is your research a necessity, what lack is there?

Response 1: (Line 78-82) Adjacent to the backdrop of these intellectual traditions, there is a need for a theoretical framework that allows Muslims to explore the purposeful complexity of the roles that narratives play in their lives. In short, I endeavour to bring to the forefront the narrative pedagogy of Islamic intellectual tradition by redressing the tracts, purposes, or maqāṣid (objectives) of the Qurʾān.

Comment 2: You mention Ricoeur, but his insights on narratives and narrativity deserve more attention, especially in the context of your research.

Response 2: Thanks for your comment. Under subchapter 3.1 Narrative as Pedagogy, I've added the following sentence,

(Line 203-208) In due course, Paul Ricoeur delved deeper into these aspects, dedicating his later endeavours to the contemplation of narrative and its significance in the realm of philosophy, including the philosophy of history and the metaphysical implications of time (Joy 1997). As Ricoeur (1991) puts it, hermeneutics represents a philosophy deeply rooted in the concept of perpetual mediatedness. It underscores that no form of self-understanding exists devoid of mediation through signs, symbols, and texts.

Comment 3: ‘It is a longstanding belief’. Please provide one of more sources that underline this claim and the sentences following (line 327-335)

Response 3: (Line 340-353) The intricate relationship between the Qurʾān and subsequent literary works is inherently nuanced and multifarious. The Qurʾān possesses the potential to not only mould literary dialogue straightforwardly but also serves as an autonomous frame of reference to rejuvenate other notions and musings. Much like other religious and literary traditions, Islamic literature showcases various customs that can be categorised into three types - those derived from the holy book through direct and rational means, those that align with its essence, and those completely unrelated to it (Bin Muhammad Yusoff and Mohd Yusuf 2023).

The narratives found within the Qurʾān possess a distinct historical context and are intricately linked three specific categories or temporal phases in history: the past, the present, and the future. Bennabi’s (2004) insights effectively encapsulate this perspective regarding narratives within the Qurʾān. Focusing on narrative literature, Roberto Tottoli (2017) retraces literary legacy and how it has been influenced, or in other words, the mark left by the Qurʾānic narratives on Islamic Arabic literature. 

Comment 4: This paragraph is nothing more than an underlining of former insights. What is there to discuss, concerning the former? Please focus on discussing issues.

Response 4: Thank you for this important comment. In subchapter 5. Discussion, I have included additional discussion as follows:

(Line 846-897) As mentioned earlier, the term “explication” is employed in an interpersonal context, rather than in a broader inferential sense. From this perspective, I propose that Muslims should perceive this shift as a transition towards a spiritually oriented intellection (Al-Attas 2014) and conceptualisation of the ontic-ontological significance of narrative pedagogy in Islamic education. This perspective sees the individual as consistently shaped through an ongoing process of narrative explication, which occurs within a pedagogical relationship that is deeply intertwined with models of sense-making mediated by self-religiosity. Narration, serving as a communicative medium, fulfils at least a dual role: elucidating textual meaning and retracing historical events. Given the Qurʾān’s guided role in revitalising and reshaping Muslims, this transition necessitates the recognition of the reciter or educator as playing a fundamental role in how listeners interpret their identities and positions within the world. Through an emphasis on interpreting and imagining as shared endeavours, interpretive pedagogies like narrative pedagogy, encourage both educators and learners to collectively harness their insights. This approach prompts them to question their preconceptions, imagine novel avenues for nurturing spirituality, and actively collaborate with others in order to promote theological reflection, spiritual growth, and social empowerment.

Hanna Meretoja (2018) argues that the process of self-reflection and self-transformation through narratives occurs when individuals engage with literary and various cultural narratives. Our lives are contemplated in connection with the narratives we encounter through hearing and recitation. Hans-Georg Gadamer (1997) underscores the notion that our self-awareness is profoundly shaped by engaging with others, especially those with differing perspectives. This recognition of the other’s standpoint becomes instrumental in revealing our inherent preconceptions, constituting a pivotal aspect of how narrative and various forms of pedagogy facilitate the process of self-examination and self-awareness. This point is in alignment with Abdallah Rothman’s (2022) recent Islamic model of the soul, where the research interest lies in developing a foundational model for Islamic approaches to psychotherapy, as well as for an Islamic theory of human psychology. When al-Ghazālī advises us that “the soul of human being is as a mirror,” he envisions his audience to find “the Real” therein (Zargar 2017). Shifting our focus in this manner enables us to recognise that the inquiry into the connection between narrative and the soul inherently encompasses the subject’s interaction with social practices and the intricate power dynamics at play.

In a broader context, the proposed framework (truth, beauty, explication) requires practical implementation in the realm of Islamic education, as well as studies that observe key concepts related to narrative, experience, subjectivity, memory, and their interrelations. The framework holds potential value for educators, policymakers, and educational authorities, particularly in Islamic countries, by highlighting the significance of narrative pedagogy in fostering theology, spirituality, and mimetic function skills among students. In the future, it may serve as a catalyst for generating innovative ideas, fresh teaching materials, and innovative pedagogical modules. Additionally, conducting ontic-ontological studies of the Qurʾānic concept would enable researchers to explore broader questions of pedagogy within Qurʾānic studies and Islamic education. This entails not only tadabbur (contemplation) of the narratives within the Qurʾān but also a comparative analysis of interpretative patterns and human soul between Qurʾānic narrative and other narrative traditions where the Qurʾān is not the central focus. How can studies of narrative contribute to cross-disciplinary dialogues within Islamic education regarding the interpretation of Qurʾānic stories, exegeses, and foundational theosophical and philosophical texts? What guidance might the framework offer educators in other, more practical fields of Islamic education, including the rapidly growing domain of digital Islamic studies? Alternatively, what insights and tracts from this presentation can educators in Islamic education glean?